# Immune Gene Repertoire of Soft Scale Insects (Hemiptera: Coccidae)

**DOI:** 10.3390/ijms25094922

**Published:** 2024-04-30

**Authors:** Andrea Becchimanzi, Rosario Nicoletti, Ilaria Di Lelio, Elia Russo

**Affiliations:** 1Department of Agricultural Sciences, University of Naples Federico II, 80126 Naples, Italy; andrea.becchimanzi@unina.it (A.B.); ilaria.dilelio@unina.it (I.D.L.); elia.russo@unina.it (E.R.); 2BAT Center—Interuniversity Center for Studies on Bioinspired Agro-Environmental Technology, University of Naples Federico II, 80126 Naples, Italy; 3Research Centre for Olive, Fruit and Citrus Crops, Council for Agricultural Research and Economics, 81100 Caserta, Italy

**Keywords:** Coccomorpha, coccoids, pathogen recognition, signaling pathways, immune response

## Abstract

Insects possess an effective immune system, which has been extensively characterized in several model species, revealing a plethora of conserved genes involved in recognition, signaling, and responses to pathogens and parasites. However, some taxonomic groups, characterized by peculiar trophic niches, such as plant-sap feeders, which are often important pests of crops and forestry ecosystems, have been largely overlooked regarding their immune gene repertoire. Here we annotated the immune genes of soft scale insects (Hemiptera: Coccidae) for which omics data are publicly available. By using immune genes of aphids and *Drosophila* to query the genome of *Ericerus pela*, as well as the transcriptomes of *Ceroplastes cirripediformis* and *Coccus* sp., we highlight the lack of peptidoglycan recognition proteins, galectins, thaumatins, and antimicrobial peptides in Coccidae. This work contributes to expanding our knowledge about the evolutionary trajectories of immune genes and offers a list of promising candidates for developing new control strategies based on the suppression of pests’ immunity through RNAi technologies.

## 1. Introduction

Due to their constant exposure to a diverse range of pathogens, parasites, and environmental stressors, insects exhibit a multi-faceted immune system [1,2]. In addition to behavioral adaptations and mechanical barriers, such as the exoskeleton, insects have evolved effective innate immunity that acts through both cellular and humoral responses directed against invaders [1,3]. Activation of these responses occurs through the recognition of pathogen-associated molecular patterns (PAMPs) by receptors, known as pattern recognition receptors (PRRs), located on hemocytes (immune cells) and epithelial cells from barrier sites throughout the insect’s body [4,5]. Cellular immune responses include phagocytosis, nodulation, encapsulation, and melanization events mediated by hemocytes [6,7,8]. Humoral responses orchestrated by signaling pathways such as Imd, Toll, Jak/Stat, and JNK lead to the synthesis of various defense enzymes, complement-like proteins, and antimicrobial peptides (AMPs) in response to infection [9,10,11]. Immune responses vary depending on the size and PAMPs of the intruder. Encapsulation and melanization function as defensive strategies against larger intruders, such as the eggs laid by endophagous parasitoids [12], whereas AMPs play a central role in mitigating the impact of pathogenic microorganisms [13,14].

Although the immune system of insects has been extensively studied, the majority of the studies on this subject are focused on holometabolous species, including the fruit fly *Drosophila melanogaster* Meigen (Diptera: Drosophilidae), honey bees, mosquitoes, beetles, and moths [15,16,17,18,19]. One of the first heterometabolous insects for which the immune system was characterized at both the genomic and functional levels was the pea aphid *Acyrthosiphon pisum* (Harris) (Hemiptera: Sternorrhyncha; Aphididae) [20,21]. Several studies indicate that this agricultural pest exhibits a reduced immune system [22], which is enhanced by its secondary bacterial symbionts [23]. The pea aphid is widely accepted as a model insect for studying a wide range of biological and physiological processes [24], including those deriving from their interaction with parasites and symbionts [25]. Furthermore, its genetic information [26] has the potential to provide valuable insights into other hemipteran pests that are less extensively studied. Indeed, recent omics studies pointed out that immune system reduction is not restricted to aphids, but is a common feature of several hemipteran species, such as *Diaphorina citri* (Hemiptera: Liviidae) [27], *Plautia stali* (Hemiptera: Pentatomidae) [28], and *Rodnius prolixus* (Hemiptera: Reduviidae) [29].

Alongside closely related hemipterans such as aphids (Aphidomorpha) and whiteflies (Aleyrodomorpha) [30], scale insects (Coccomorpha) are sap-sucking and obligate plant parasites [31,32]. The feeding behavior of scale insects delays plant growth and, in severe infestations, can lead to the death of the entire plant [31]. Indirect damage derives from the production of honeydew, which results in the growth of saprophytic fungi, thereby reducing the rate of plant photosynthesis and causing decline [31]. In addition, some species may act as vectors of pathogenic viruses [33]. This taxonomic group shows remarkable diversity in both external and internal morphology, as well as in reproduction and symbiotic systems, making them a fascinating subject for scientific study [31]. Scale insects, whose name derives from the commonly produced protective covering (“scale”), exhibit peculiar adaptations. They have sexual dimorphism, characterized by ephemeral alate males lacking functional mouthparts and stationary non-winged adult females, which produce a variety of protective waxy secretions [31].

The majority of these insects, owing to their imbalanced diet rich in carbon, engage in obligate symbiotic relationships with different species of bacteria or fungi [34,35,36]. Some species exclusively harbor a single obligate symbiont responsible for the synthesis of essential nutrients [37,38], while others form additional associations with facultative symbiotic organisms [39,40]. Symbiotic microorganisms are housed in specialized cells known as bacteriocytes (or mycetocytes), fat body cells, or the midgut epithelium or are dispersed in the hemolymph [34,41].

Among scale insects, the family Coccidae, commonly known as “soft scale insects”, includes approximately 1180 species worldwide [42]. Some of these are important pests of crops [43] or forest plants [44]. These pests have a diverse complex of natural enemies that can control their populations, including commercially available predatory insects and naturally occurring parasitoids [45]. The most important group of their antagonists comprises chalcidoid wasps (Hymenoptera: Chalcidoidea) [46,47], which mainly belong to the Encyrtidae, Aphelinidae, and Eulophidae families [46]. Ladybirds (Coleoptera: Coccinellidae), particularly *Cryptolaemus montrouzieri* and *Chilocorus* sp., are well-known predators of Coccidae and have proven effective in various biological control programs [48,49,50]. Furthermore, a number of entomopathogenic fungi can infect and exert a detrimental effect on scale insects. Besides the best known entomopathogens, such as species of *Akanthomyces/Lecanicillium*, which are commonly reported to haunt populations of Coccidae [51,52], new species have been characterized in recent years for their pathogenicity to these pests [53,54,55,56]. Moreover, species in certain fungal genera that are frequently reported to establish an endophytic association with plants, such as *Fusarium* and *Cladosporium*, could play an ecological role in the containment of scale insects [51,57,58]. However, the most intriguing relationship concerns species of *Ophiocordyceps*. Also referred to with the anamorphic name *Hirsutella*, these fungi are commonly regarded as specialized entomopathogens [59]. Notwithstanding, evidence from several independent studies has demonstrated that they develop an intimate symbiotic relationship with soft scales involving their transovarial transmission between generations [35,60,61], which deserves to be examined more in depth in view of possible applications of its disruption in pest control.

As with most sternorrhynchan species, the immunity of soft scale insects has not been thoroughly investigated and remains to be clarified. To date, only a few studies have been conducted on the immune response of Coccidae, mainly aimed at determining the encapsulation response against eggs released by encyrtid parasitoid wasps [62,63,64]. The limited understanding of soft scales’ immunity hinders our comprehension of their interactions with symbionts and pathogens, as well as the potential development of new control strategies based, for example, on the suppression of the immune response through RNA interference, as recently proposed [65,66].

In the present work, we aimed to identify and annotate immune genes in soft scale insects by searching the recently published genome of the Chinese white wax scale insect (*Ericerus pela*) (Chavannes), as well as the transcriptomes of *Coccus* sp. and *Ceroplastes cirripediformis* Comstock (Hemiptera: Coccidae).

## 2. Results and Discussion

### 2.1. Overview of Immune Genes’ Annotation

We focused our annotation efforts on a subset of genes involved in the three phases of the insect immune response: recognition, signaling, and response. All annotations are based on the recently completed sequencing of *E. pela* (colony RIRI-1) [67]. By using protein sequences from *A. pisum* and *D. melanogaster*, known to be involved in insect immunity, as queries in BLAST searches, we successfully identified 66 potential immune genes in *E. pela*. Specifically, we identified 12 genes related to recognition, 35 involved in signaling, and 19 associated with the response to pathogenic microorganisms (Figure 1).

### 2.2. Annotation of Recognition Genes

Our analysis pointed out the occurrence of 12 genes in Coccidae with significant matches with *A. pisum* and *Drosophila* genes involved in recognition (Table 1). As occurs in aphids, Coccidae species lack peptidoglycan recognition proteins (PGRPs), class C scavengers, and Nimrod and eater receptors.

#### 2.2.1. Peptidoglycan Receptor Proteins

Peptidoglycans are essential cell wall components of almost all bacteria, which are recognized by the immune system through pathogen recognition receptors (PRRs). In insects, several families of pattern recognition molecules that detect peptidoglycans have been identified, and the role of peptidoglycan receptor proteins (PGRPs) in host defense is relatively well-characterized in *Drosophila* [68]. PGRP-based recognition activates both the Toll and IMD/JNK pathways, leading to proPO activation or the synthesis of antimicrobial peptides [69]. Most insect species investigated possess several *PGRP* genes that differ both structurally and functionally. For example, *Drosophila* has 13 PGRP genes encoding 19 proteins, while *Anopheles gambiae* has 7 PGRP genes encoding 9 proteins [68]. However, like the pea aphid [22], Coccidae appear to have no PGRPs (Table 1).

#### 2.2.2. Gram-Negative Binding Proteins

Gram-negative binding proteins’ (GNBPs) architectures consist of a carbohydrate-binding module (CBM) at the N-terminus and a glucanase-like domain (Glu) in the C-terminus [70]. The CBM interacts with microbial polysaccharides, while the Glu domain interacts with downstream proteases, thereby initiating immune pathways [71]. GNBPs recognize both bacterial and fungal pathogens, resulting in the activation of immune signaling pathways in insects [72]. Specifically, in *Drosophila*, GNBP1 and peptidoglycan-recognition protein-SA (PGRP-SA) collaboratively activate the Toll pathway in response to Gram-positive bacterial infections [73], whereas GNBP3 is essential for Toll pathway activation in response to fungal infections [74].

A study based on gene knockdown revealed that the two GNBPs predicted in the genome database of pea aphids [22] are involved in the antibacterial response in the pea aphid, likely acting as PRRs in the prophenoloxidase pathway [75]. Our analysis identified a single gene encoded in the transcriptome and the genome of *C. cirripediformis* and *E. pela*, respectively, while two different genes seem to occur in the transcriptome of *Coccus* sp. (Figure 2).

#### 2.2.3. Lectins

Lectins, a diverse group of sugar-binding proteins, are integral to the immune response of several insect species. They are known for their broad spectrum of pathogen binding and involvement in various immune processes such as opsonization, melanization, antibacterial peptide synthesis, encapsulation, and direct killing of bacteria [76]. *Drosophila* c-type lectins (CTLs) have been implicated in facilitating the encapsulation of parasitoid invaders by marking surfaces for hemocyte recruitment [77]. Interestingly, as in *A. pisum*, in Coccidae, no homologs of *D. melanogaster* DL1 (AAF53793.1) have been found (Table 1).

Our phylogenetic reconstruction (Figure 3) revealed that the two CTLs identified in Coccidae are more closely related to DL2 (NP_001014489.1) than to DL3 (NP_001014490.1). This suggests that the phylogeny of CTLs is characterized by species-specific contraction and expansion events, influenced by factors such as environmental pressure, pathogen interactions, and microbiota [78].

Galectins, another widely distributed group of lectins [79], are upregulated in mosquitoes in response to both bacterial and malaria parasite infection [80]. Insect galectins are thought to be involved in pathogen recognition, agglutination, and phagocytosis [79,81]. Genome-wide analyses have revealed variation in galectin transcripts across insect species, with 5 in *D. melanogaster*, 8 in *A. gambiae*, 12 in *Aedes aegypti* [82], 4 in many Lepidoptera species [83], and 1 in aphids. In contrast, Coccidae lack *galectin* putative homologs (Table 1).

#### 2.2.4. Thioester-Containing Proteins

Thioester-containing proteins (TEPs) are a family of proteins structurally related to vertebrate complement proteins, including an intramolecular β-cysteinyl-γ-glutamyl thioester bond [84,85]. As observed in complement proteins, some TEPs are involved in the opsonization of microbes and pathogens, ‘marking’ them for phagocytosis, melanization, and the formation of lytic complexes [86,87]. Due to their involvement in microbe recognition, TEPs can be classified as PRRs [85].

As in aphids, Coccidae omics data showed the presence of one TEP gene encoding two isoforms, except in *Coccus* sp., where only one isoform was found (Figure 4). The most closely related proteins are the two isoforms encoded by the only TEP ortholog in the *A. pisum* genome. Indeed, in contrast to what is reported in [22], referring to an old annotation, only one TEP ortholog was identified by our analysis in the *A. pisum* genome using four *Drosophila* homologs (*TepI*, *TepII*, *TepIII*, and *TepIV*) as the query (Table 1).

#### 2.2.5. Class C Scavenger and Nimrod Receptors

As observed in pea aphids, both class C scavenger and Nimrod receptors are absent in Coccidae (Table 1). Class C scavenger receptors, which have been identified only in *Drosophila*, exhibit a broad affinity toward both Gram-positive and Gram-negative bacteria [88].

The Nimrod family of proteins is characterized by the presence of epidermal growth factor (EGF)-like domains, also called ‘NIM repeats’ [89]. Several members of the Nimrod superfamily appear to function as receptors in phagocytosis and bacterial binding [90,91]. NimC1 and Eater, two EGF-like repeat Nimrod surface receptors specifically expressed in hemocytes, synergistically contribute to bacterial phagocytosis [92] and are both absent in Coccidae.

### 2.3. Annotation of Signaling Pathways

Our analysis revealed the occurrence of 35 genes in Coccidae with significant matches with genes of *Drosophila* and *A. pisum* involved in signaling. Coccidae lack MyD88, TNF-receptor-associated factor 3, and cactus genes, belonging to the Toll pathway, as well as several members of IMD signaling pathway, which are present in *Drosophila* and *A. pisum* genomes (Table 2).

#### 2.3.1. The Toll Signaling Pathway

The Toll pathway in *Drosophila* functions in both development and innate immunity. Deletion of its component genes increases the susceptibility to various pathogens, including Gram-positive bacteria, fungal pathogens, some Gram-negative bacteria, and viruses [93]. Moreover, upregulation of Toll pathway components occurs in response to parasitoid wasp invasion [94]. The Toll pathway appears to be intact in Coccidae, except the MyD88 adaptor and the inhibitor molecule cactus (a homolog of IkB) (Table 2), which are instead present in *A. pisum*. We found convincing matches for genes encoding the extracellular cytokine *spätzle*, the transmembrane receptor Toll (Figure 4), the tube adaptor, the kinase pelle, cactin, pellino, Traf, and the transactivator dorsal (Table 2). Coccidae seem to have multiple *spätzles,* putative homologs of *Drosophila spätzles* 1, 2, 3, 4, and 6 (Table 2), for which a phylogenetic reconstruction was not possible due to high divergence between the different *spätzle* subfamilies [95].

Coccidae also have multiple genes encoding Toll receptors (Figure 5), which function as transmembrane receptors in both mammals and insects. While nine single-copy Toll genes have been identified in *D. melanogaster* (*Toll1* to *Toll9*), it seems that Coccidae, like other insects, lack some of these genes, but have multiple isoforms of others. Notably, no Toll6 and Toll2/7 homologs have been found in the *C. cirripediformis* transcriptome, while *Coccus* sp. lacks Toll10 homologs (Figure 5).

In other organisms, some Toll subfamilies are involved in immune function, while others function in developmental processes [96]. However, an accurate homology-based approach including different species is essential for understanding Toll functions in Coccidae.

#### 2.3.2. The JAK/STAT Signaling Pathway

In *Drosophila,* the JAK/STAT pathway, similar to the Toll pathway, plays roles in both development and immunity. Despite being the least understood of the core insect immune pathways, it appears to induce hemocyte overproliferation and antiviral responses [97]. Additionally, changes in gene expression observed after parasitoid wasp invasion of *Drosophila* larvae indicate the involvement of the JAK/STAT pathway in the response to parasitoids [98].

As in *A. pisum*, Coccidae have homologs of all core JAK/STAT genes, including genes encoding the cytokine receptor domeless (Figure 6), JAK tyrosine kinase (also known as Hopscotch), and the STAT92E transcription factor (Table 2). However, no homologs were found for *upd* (unpaired), considered a key ligand in *Drosophila* JAK/STAT induction. This ligand is also missing in other insects (e.g., *A. mellifera*) [99]. The presence of the core JAK/STAT pathway members (Table 2) suggests that JAK/STAT remains functional in Coccidae and is triggered by a currently unrecognized ligand.

#### 2.3.3. IMD and JNK Signaling Pathways

The IMD pathway is critical for fighting Gram-negative bacteria in *Drosophila* [93], and IMD pathway member knockouts influence the susceptibility to some Gram-positive bacteria and fungi as well [100]. As in *A. pisum*, Coccidae appear to be missing many crucial components of the IMD signaling pathway, such as *IMD*, *dFADD*, *kenny*, and *Relish* (*Rel*) (Table 2).

Pea aphids lack genes associated with the IMD pathway but possess orthologs for most components of the JNK pathway. In *Drosophila*, the JNK pathway is involved in various developmental processes, along with wound healing, and has been suggested to regulate antimicrobial peptide gene expression and cellular immune responses [93]. The genes involved include *hep*, *basket*, and *JRA*. Searches for homologs to the *Drosophila kayak* (*kay*) gene found no hits in Coccidae. Considering that this gene is also involved in controlling viral infections [101], understanding its role in the Coccidae family, whose members are known for their capacity to transmit plant viruses [33], may be of importance in the management of vector-borne plant pathogens.

Given that in *Drosophila*, the IMD pathway activates components of the JNK pathway [93], the presence of JNK but absence of the IMD signaling pathway in Coccidae suggests that an alternative pathway for JNK activation, independent of IMD, involving the inducer Eiger, may occur [22].

### 2.4. Annotation of Response Genes

Our analysis revealed the occurrence of 19 genes in Coccidae with significant matches with genes of *Drosophila* and *A. pisum* involved in the immune response. As occurs in aphids, Coccidae species lack antimicrobial peptides; moreover, they do not possess thaumathins, which are present in the *A. pisum* genome (Table 3).

#### 2.4.1. Antimicrobial Peptides

Antimicrobial peptides (AMPs) play a key role in the immune response of many organisms, including insects. They are the most widely studied humoral effectors and can be produced constitutively or following induction through particular signaling pathways. In the *Drosophila* genome, there are currently seven well-characterized families of inducible AMPs, including 21 AMP/AMP-like genes, which play an important role both in counteracting the onset of infections and in maintaining homeostasis with symbiotic microorganisms [102]. The main site of production of these molecules is represented by fat body cells; however, hemocytes and cells of the cuticular epithelium, intestinal epithelium, and reproductive tract are also involved in the synthesis of AMPs. Most AMPs are 15–50 amino acids long and have an amphipathic structure; they are able to alter the permeability of the microbial membrane, generating an alteration of the osmotic balance, with consequent lysis [10]. Currently, several structural families of AMPs from insects are known (defensins, cecropins, drosocins, attacins, diptericins, metchnikowins, and melittins), some of which are peculiar to a particular taxonomic group. For example, drosomycins have only been identified in *Drosophila*, while gloverins have been specifically described in Lepidoptera; cecropins, and attacins, meanwhile, are known in several insect species, while defensins are widely distributed throughout the animal kingdom [14].

As observed in pea aphids [22], Coccidae are missing many of the antimicrobial peptides common to other insects. Extensive searches for genes encoding AMPs previously identified in Hemiptera species (thanatin, pentatomicin, lugensin, cicadin, cryptonin, pyrrhocoricin, oncocin, hemiptericin) also revealed no hits (Appendix A). Even the six *thaumatin* homologs in the *A. pisum* genome, which show overall sequence and predicted structural similarities to plant thaumatins [22], are absent in Coccidae (Table 3). Recently, it was suggested that the impact of selection on the innate immune system can act on AMPs, indicating that some AMPs can be deleterious molecules in the absence of microbial challenges, due to their costly production and/or their toxic effects [102]. We speculate that such AMP loss derives from a reduced pathogen pressure in Coccidae and other hemipterans, which feed on the generally microbe-free plant phloem [103].

#### 2.4.2. Lysozyme and Peptidoglycan Degradation

Lysozymes, enzymes responsible for breaking down bacterial cell walls by targeting the polysaccharide component of peptidoglycan, are classified into two classes in insects: the c-type, with muramidase activity, and the i-type, which possess both muramidase and isopeptidase activities [104,105]. As observed in *A. pisum* [22], Coccidae lack genes for several lysozymes (*LysB*, *LysD*, *LysE*, and *LysP*), which are highly expressed in the gut of *Drosophila* and are involved in regulating the microbial composition and in degrading peptidoglycan from dietary bacteria [106].

Only three genes encoding i-type Lys were identified in the genome of *E. pela* (Table 3) and in the transcriptome of *Coccus* sp., while there were only two in the transcriptome of *C. cirripediformis*. The i-type Lys sequences of Coccidae formed three monophyletic groups: two contain an invertebrate (I)-type lysozyme domain profile (LYSOZYME_I) and EF-hand calcium-binding domain, and one contains the LYSOZYME_I domain alone (Figure 7). Notably, *C. cirripediformis* lacks the lysozyme with the LYSOZYME_I domain alone.

Although numerous insect genes encoding both c-type and i-type lysozymes have been identified through genome and transcriptome analyses, i-type lysozymes have been poorly investigated from a functional perspective [107]. An i-type lysozyme of the beetle *Harmonia axyridis* was recombinantly expressed in the yeast *Pichia pastoris*, but the purified protein showed no muramidase and no isopeptidase activity [108]. Transcription and immunofluorescence analysis revealed that this i-type lysozyme is produced in the fat body cells but is not inducible by immune challenge. These findings suggest that i-type lysozymes in insects may have acquired novel and as yet undetermined functions during evolution [108].

One of the defining characteristics of the Hemiptera biology is their mutualistic symbiosis with microorganisms. Symbiotic bacteria and fungi play crucial roles in tasks such as nutrition and defense, often residing within specialized cells (bacteriocytes) or dispersed throughout the hemolymph [35,109].

In the bacteriocytes of *A. pisum*, two genes thought to be involved in the degradation and recycling of peptidoglycan, LD carboxypeptidase (ldcA) and rare lipoprotein A (rlpA), are expressed at high levels [110]. These genes have been acquired from bacteria of the genus *Wolbachia* or *Rickettsia* through horizontal gene transfer [111,112,113].

Our analysis did not yield any significant matches for rlpA, but for ldcA, we found a single hit in the genome of *E. pela*. However, our phylogenetic reconstruction revealed that the protein is encoded by ldcA clusters for *Rickettsia* and is poorly related with sequences found in aphid genomes (Figure 8). This suggests that the identified protein is likely encoded by the genome of the bacterial symbiont *Rickettsia* sp., known to colonize the gut of *E. pela* [114]. The presence of this sequence in the assembled genome of *E. pela* is likely due to contamination of the sample with DNA from the gut symbiont. The lack of ldcA acquisition by the *E. pela* genome is in line with previous studies on symbiotic microorganisms of Coccidae, which, rather than being intracellular bacteria as in aphids, are mostly fungi localized in hemolymph, fat body cells, and ovarioles [36].

Moreover, we assessed the presence of homologs of bacteriocyte-specific cysteine-rich (BCR) peptides, which are small disulfide bond-rich proteins expressed exclusively in aphid bacteriocytes putatively involved in endosymbionts’ control and belonging to a structural class of defensins [115,116]. However, our analysis did not yield any significant matches, supporting the hypothesis that Coccidae lack bacteriocytes and their specific mechanisms of symbiosis mediation.

#### 2.4.3. Chitinases

Chitinases, which are glycosyl hydrolases, are enzymes responsible for breaking down chitin, a polymer of N-acetyl-d-glucosamine and the second most abundant biopolymer worldwide. Chitin serves as a structural component in various biological matrices, such as arthropod exoskeletons and fungal cell walls [117]. In arthropods, chitinases fulfill dual roles: aiding in molting processes and serving as defense mechanisms against parasites like fungi and nematodes [118,119]. These enzymes target chitin by hydrolyzing the 1,4-β-linkages between its constituent glucosamine units. Chitinases and lysozymes belong to the same superfamily of hydrolases, exhibiting similar catalytic activities. In fact, certain chitinases possess lysozyme activity, and vice versa [120].

Chitinase-like proteins of Coccidae are included in the majority of the groups classified on a phylogenetic basis [118], except for groups IV and VII where no Coccidae sequences have been identified. In group II, the only member is from *C. cirripediformis*, while group I includes only sequences from *E. pela* and *Coccus* sp. (Figure 9). Further studies are required to determine the biochemical properties and enzymatic activities of these chitinase-like proteins in Coccidae.

#### 2.4.4. Prophenoloxidase

Phenoloxidase-mediated melanin formation is a characteristic feature accompanying wound clotting, phagocytosis, and encapsulation of pathogens and parasites [122]. In aphids, the inactive enzyme prophenoloxidase (ProPO) is activated by serine proteases to produce phenoloxidase [123,124]. ProPO-activating proteinases, such as phenoloxidase activating factor 2 (paf2) in *A. pisum*, are upregulated in response to parasitization by parasitic wasps [125]. Coccidae appear to possess more than one ProPO homolog, as observed in *A. pisum*.

Furthermore, our analysis revealed the presence of two paralogs belonging to the paf2 family in the considered species of Coccidae. The sequences from the three Coccidae species form two monophyletic groups, with the most closely related sequences being their orthologs in the *A. pisum* genome (Figure 10).

#### 2.4.5. Nitric Oxide Synthase

Nitric oxide is a highly unstable free radical gas that has been shown to be toxic to both parasites and pathogens [126]. Production of nitric oxide is mediated by the enzyme nitric oxide synthase (Nos) and triggers the activation of the Toll/IMD signal pathway [127]. In insects, *Nos* is upregulated after both parasite [128] and bacterial infection [129]. Like pea aphids, *E. pela* have one *Nos* homolog (Table 3).

## 3. Materials and Methods

Immune gene candidates from *A. pisum,* identified by [22] and reported in Table 1, Table 2 and Table 3, were used to query the *E. pela* genome (GenBank: GCA_011428145.1). Most searches utilized the tblastn search to search for hits against the assembled *E. pela* genome, considering as positive only the hits with an e-value less than 1 × 10^−5^. For genes absent in the *A. pisum* genome, genes from *D. melanogaster* were used to query the *E. pela* genome.

To identify immune-related genes in Coccidae, we aligned the *A. pisum* major immune protein sequences on the *E. pela* genome, using *exonerate* [130], and we extracted the translated CDS, using *getorf* [131]. In brief, the protein sequences of *A. pisum* immune genes were used as inputs in *exonerate* using the protein2genome model, which allows introns in the alignment, but also allows frameshifts, and exon phase changes when a codon is split by an intron [130]. The resulting sequences were translated using *getorf*, which finds and outputs the sequences of open reading frames (ORFs) in nucleotide sequences [131].

To reconstruct the immune gene phylogeny of Coccidae, the identified immune-related protein sequences of *E. pela* were used as a query in tblastn searches against the assembled transcriptome of *C. cirripediformis* (TSA project accession: GCWZ01) and *Coccus* sp. (TSA project accession: GCWW01). Putative homologous sequences in other insect species (*Apis mellifera*, *Tribolium castaneum*, and *Bombyx mori*) were identified by sequence similarity searches through BlastP, using *D. melanogaster* proteins (Table 1, Table 2 and Table 3) as the query versus the non-redundant NCBI database (nr NCBI db). However, using the *ldcA* protein sequence as a query resulted in no hits in the above-mentioned species. Therefore, a blastP search against the whole nr NCBI db was also performed to reconstruct the *ldcA* phylogeny. One best hit per query was selected and all the protein sequences were aligned using Muscle 3.8 [132], with default settings.

Alignments were automatically trimmed using Gblocks version 0.91b [133] to avoid comparisons of non-conserved regions present only in a subset of the taxa. The best-fit model of amino acid substitution and phylogenetic reconstruction was generated using RAxML 8.2.12 [134]. The maximum likelihood tree was run for 1000 bootstrap replicates and the tree figure was plotted using FigTree v1.4.3. Protein sequences were analyzed with ScanProsite (https://prosite.expasy.org/scanprosite/, accessed on 5 March 2024) in order to identify active sites and conserved patterns [135].

## 4. Conclusions

Our annotation of Coccidae immune genes sheds light on the poorly explored repertoire of defense molecules and mechanisms used by a group of insects with a peculiar morphology and habits. The immune gene loss pattern recalls what is observed in aphids, with some exceptions (i.e., genes that are absent in Coccidae, but present in aphids). These are (1) galectins, involved in recognition; (2) some members of the Toll and JNK pathways; and (3) thaumatins, which are antimicrobial effectors also identified in Coleoptera and even in plants. Understanding if this erosion of the immune repertoire is the result of a reduced pressure due to the fungal endosymbiont presence or sterile lifestyles, such as plant-sap feeding, requires further investigations.

Our approach consisted of running searches using the immune gene repertoire of aphids as the query against Coccidae genomics and transcriptomics assemblies. Indeed, aphids are one of the closest relatives of Coccidae, which have a deeply characterized immune gene repertoire. This approach has the advantage of being robust, although somewhat conservative, because it does not take into account alternative pathways and uncharacterized genes that may have evolved in Coccidae (and scale insects, in general), also as an adaptation to the observed loss of immune genes.

However, this work represents the first overview of the immune gene diversity of Coccidae, which we hope will inspire future studies aimed at functionally characterizing the identified genes. Indeed, understanding the role of Coccidae immune genes is a key step for developing new strategies of pest management based on the suppression of the immune response, to enhance the killing activity of entomopathogens.

## Figures and Tables

**Figure 1 ijms-25-04922-f001:**
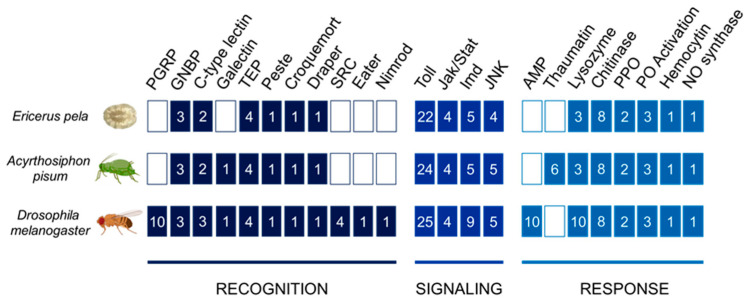
Overview of gene families involved in recognition, signaling, and response pathways in *Ericerus pela*, *Acyrthosiphon pisum*, and *Drosophila melanogaster*. Numbers in blocks indicate the different genes identified for each protein family. Abbreviations: peptidoglycan recognition (PGRP), Gram-negative binding (GNBP), thioester-containing (TEP), scavenger receptor class C (SRC), antimicrobial peptide (AMP), pro-phenoloxidase (PPO), phenoloxidase (PO), nitric oxide (NO).

**Figure 2 ijms-25-04922-f002:**
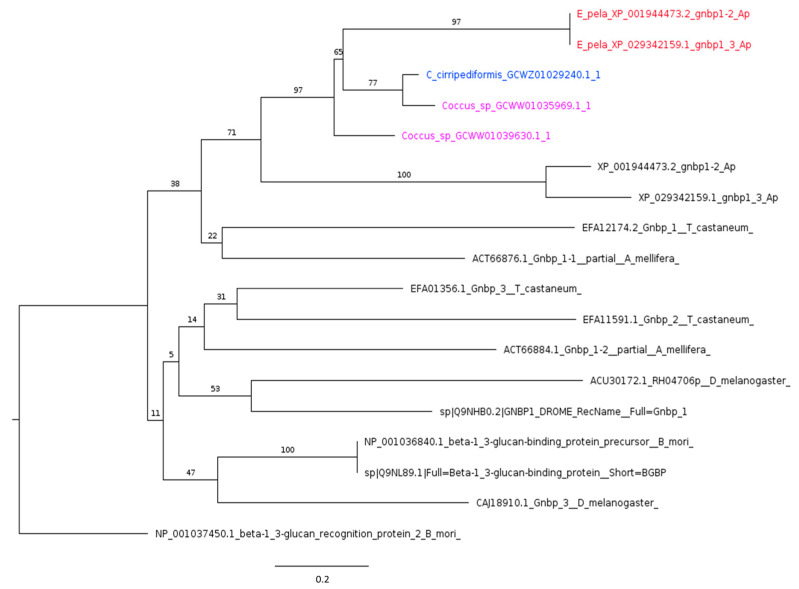
Phylogenetic tree based on maximum likelihood analysis of GNBP1 putative homologs identified in *E. pela* genome (in red) and in the transcriptomes of *C. cirripediformis* (in blue) and *Coccus* sp. (in purple). GNBP1 sequences of Coccidae family form a monophyletic group, closely related to GNBP1 of *A. pisum.* Sequences of *E. pela* are two isoforms encoded by the same gene, while *Coccus* sp. shows two different genes encoding homologs of GNBP1. One of these proteins, GCWW01035969.1, is more closely related to the GNBP1 identified in *C. cirripediformis*. The longest branch of the unrooted tree is used as the outgroup. Bootstrap support values are indicated at each node. The scale bar indicates the number of amino acid substitutions per site.

**Figure 3 ijms-25-04922-f003:**
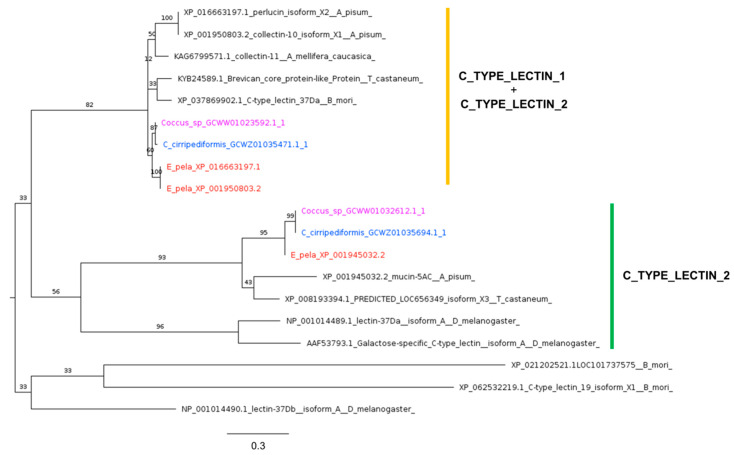
Phylogenetic tree based on maximum likelihood analysis of CTL putative homologs identified in *E. pela* genome (in red) and in the transcriptomes of *C. cirripediformis* (in blue) and *Coccus* sp. (in purple). CTL sequences of Coccidae family form two monophyletic groups, one with c-type domain signature (C_TYPE_LECTIN_1) and profile (C_TYPE_LECTIN_2), and another with c-type domain profile alone (C_TYPE_LECTIN_2), marked in yellow and green, respectively. Sequences of *E. pela* in the yellow group are two isoforms encoded by the same gene. The longest branch of the unrooted tree (DL3) is used as the outgroup. Bootstrap support values are indicated at each node. The scale bar indicates the number of amino acid substitutions per site.

**Figure 4 ijms-25-04922-f004:**
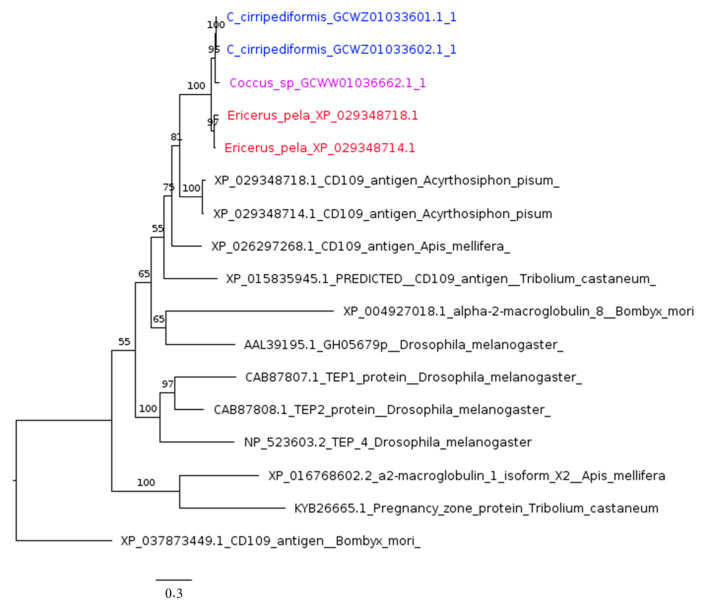
Phylogenetic tree based on maximum likelihood analysis of thioester-containing protein (TEP) putative homologs identified in genome of *E. pela* (in red) and in the transcriptomes of *C. cirripediformis* (in blue) and *Coccus* sp. (in purple). Sequences of *C. cirripediformis* are two isoforms encoded by the same gene. The longest branch of the unrooted tree is used as the outgroup. Bootstrap support values are indicated at each node. The scale bar indicates the number of amino acid substitutions per site.

**Figure 5 ijms-25-04922-f005:**
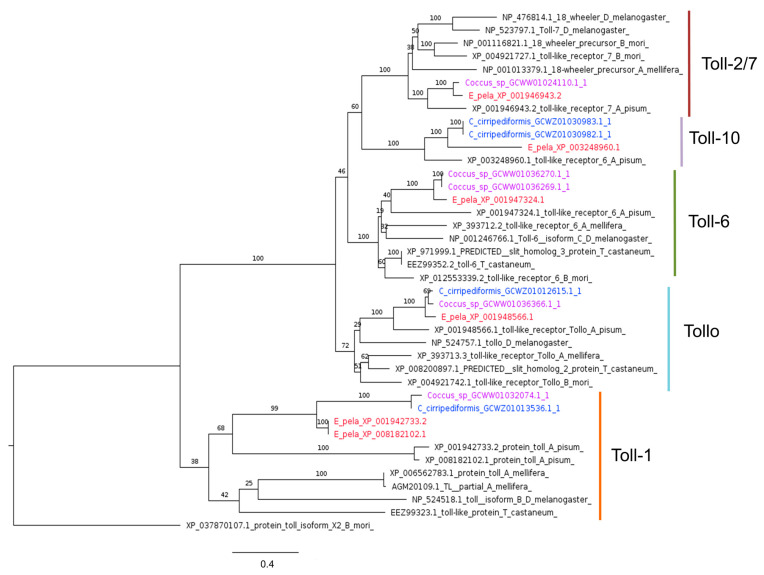
Phylogenetic tree based on maximum likelihood analysis of Toll-1, Toll-6, Toll-7, Toll-10, and Tollo putative homologs identified in *E. pela* genome (in red) and in the transcriptomes of *C. cirripediformis* (in blue) and *Coccus* sp. (in purple). *E. pela* sequences in the Toll-1 group are two isoforms encoded by the same gene. The *A. pisum* sequence XP_003248960.1, which is annotated as the Toll-6 receptor, is here annotated as Toll-10 following the classification of [95]. The longest branch of the unrooted tree is used as the outgroup. Bootstrap support values are indicated at each node. The scale bar indicates the number of amino acid substitutions per site.

**Figure 6 ijms-25-04922-f006:**
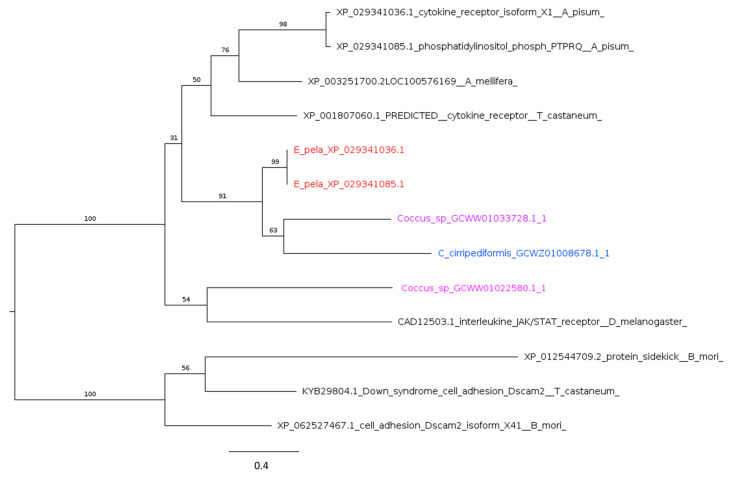
Phylogenetic tree based on maximum likelihood analysis of interleukin JAK/STAT receptor (domeless) putative homologs identified in *E. pela* genome (in red) and in the transcriptomes of *C. cirripediformis* (in blue) and *Coccus* sp. (in purple). Sequences of *E. pela* are two isoforms encoded by the same gene. The closely related dscam2 (Down syndrome cell adhesion) sequences are used as the outgroup. Bootstrap support values are indicated at each node. The scale bar indicates the number of amino acid substitutions per site.

**Figure 7 ijms-25-04922-f007:**
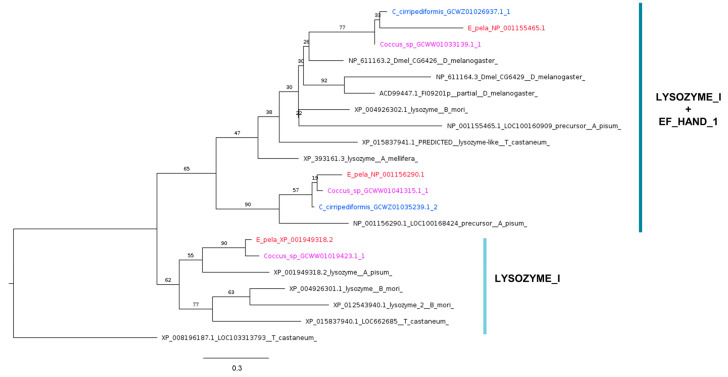
Phylogenetic tree based on maximum likelihood analysis of lysozyme putative homologs identified in *E. pela* genome (in red) and in the transcriptomes of *C. cirripediformis* (in blue) and *Coccus* sp. (in purple). Lysozyme sequences of Coccidae family form three monophyletic groups, one with the invertebrate (I)-type lysozyme domain profile (LYSOZYME_I) alone, and another with LYSOZYME_I and EF-hand calcium-binding domain (EF_HAND_1), marked in light and dark blue, respectively. The longest branch of the unrooted tree is used as the outgroup. Bootstrap support values are indicated at each node. The scale bar indicates the number of amino acid substitutions per site.

**Figure 8 ijms-25-04922-f008:**
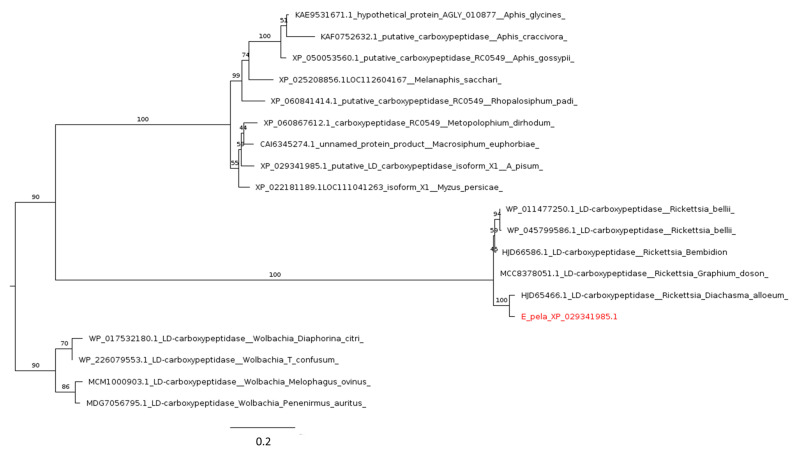
Phylogenetic tree based on maximum likelihood analysis of ldca (LD carboxypeptidase) putative homologs identified in *E. pela* genome (in red). The sequence of *E. pela* groups with LD carboxypeptidases of *Rickettsia* sp., which is a bacterial symbiont of several insect species, including *E. pela*. Sequences identified in *Wolbachia* species are used as the outgroup. Bootstrap support values are indicated at each node. The scale bar indicates the number of amino acid substitutions per site.

**Figure 9 ijms-25-04922-f009:**
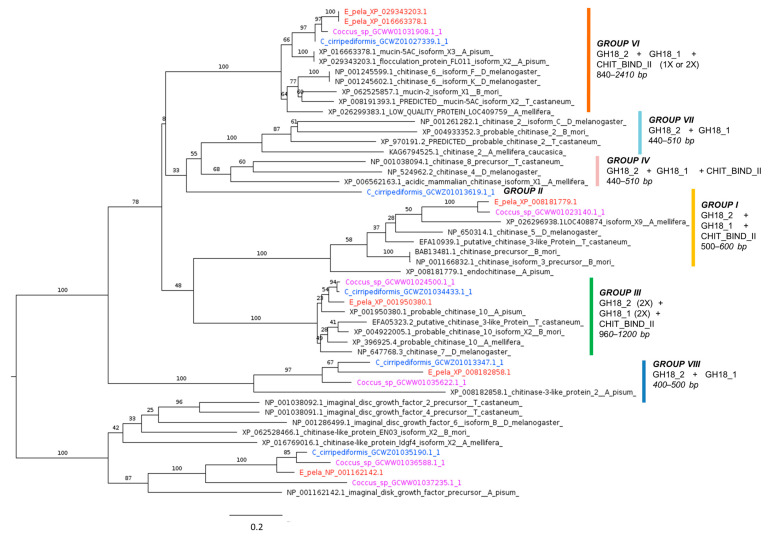
Phylogenetic tree based on maximum likelihood analysis of chitinase putative homologs identified in the *E. pela* genome (in red) and in the transcriptomes of *C. cirripediformis* (in blue) and *Coccus* sp. (in purple). Chitinase sequences of the Coccidae family are included in all the groups identified in [118], except for groups IV and VII. Chitinase groups are marked by a vertical colored bar, along with their domain architecture and length, except for sequence GCWZ01013619.1_1 of *C. cirripediformis*, which is the only member identified in Coccidae belonging to group II. Domain architecture identified by ScanProsite consists of the glycosyl hydrolases family 18 (GH18) domain (GH18_2) and active site (GH18_1) and chitin-binding type-2 domain (CHIT_BIND_II). Imaginal disk growth factor (group V) chitinase-like proteins, involved in morphogenesis and CO_2_ response rather than immunity [121], are used as the outgroup. Bootstrap support values are indicated at each node. The scale bar indicates the number of amino acid substitutions per site.

**Figure 10 ijms-25-04922-f010:**
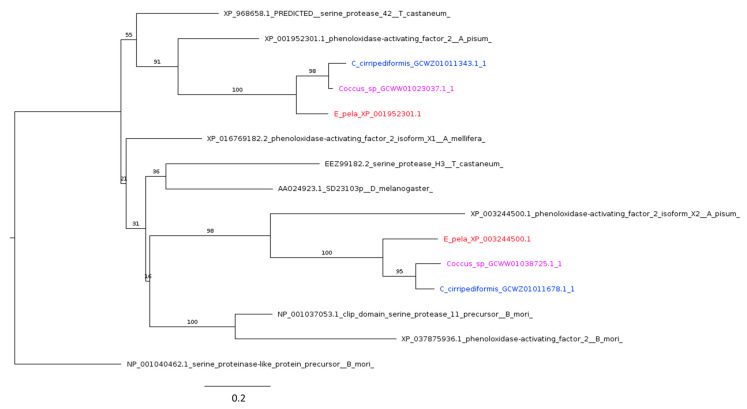
Phylogenetic tree based on maximum likelihood analysis of phenoloxidase activating factor 2 (paf2) putative homologs identified in the *E. pela* genome (in red) and in the transcriptomes of *C. cirripediformis* (in blue) and *Coccus* sp. (in purple). Each Coccidae species has two paralogs belonging to the paf2 family. The sequences of the three Coccidae species form two monophyletic groups, whose most closely related sequences are their orthologs in the *A. pisum* genome. The longest branch of the unrooted tree is used as the outgroup. Bootstrap support values are indicated at each node. The scale bar indicates the number of amino acid substitutions per site.

**Table 1 ijms-25-04922-t001:** Immune genes of *Ericerus pela* involved in recognition. Genes not found in *E. pela* are colored in red.

Role in Insect Immunity	Gene Symbol	Gene Name ^1^	*D. melanogaster* NCBI Protein ID	*A. pisum* NCBI Protein ID	Best Matches (E-Value) ^2^
bacterial recognition	PGRP-LC	peptidoglycan recognition protein	AAF50302.3	not found	not found
activation of PPO cascade and autophagy	PGRP-LE	peptidoglycan recognition protein	NP_573078.1	not found	not found
bacterial recognition	PGRP-SA	peptidoglycan recognition protein	AAF48056.1	not found	not found
bacterial recognition	PGRP-SD	peptidoglycan recognition protein	CAD89193.1	not found	not found
bacterial recognition	PGRP-LB	peptidoglycan recognition protein	NP_650079.1	not found	not found
bacterial recognition	PGRP-SC1a	peptidoglycan recognition protein	CAD89161.1	not found	not found
bacterial recognition	PGRP-SC2	peptidoglycan recognition protein	CAD89187.1	not found	not found
pgn degradation and antibacterial activity	PGRP-SB1	peptidoglycan recognition protein	CAD89136.1	not found	not found
blocking of imd pathway	PGRP-LF	peptidoglycan recognition protein	NP_648299.3	not found	not found
activation of imd pathway	PGRP-LA	peptidoglycan recognition protein	AAF50304.2	not found	not found
bacterial and fungal pattern recognition	GNBP1	Gram-negative binding protein 1	Q9NHB0.2	XP_001944473.2	QBOQ01000878.1 (3 × 10^−8^) QBOQ01000589.1 (2 × 10^−6^)
bacterial and fungal pattern recognition	GNBP2	Gram-negative binding protein 2	ACU30172.1	XP_001944473.2	QBOQ01000878.1 (3 × 10^−8^) QBOQ01000589.1 (2 × 10^−6^)
bacterial and fungal pattern recognition	GNBP3	Gram-negative binding protein 3	CAJ18910.1	XP_029342159.1	QBOQ01000878.1 (1 × 10^−18^)
bacterial recognition, induction of PPO cascade	DL1	c-type lectin 1	AAF53793.1	not found	not found
bacterial recognition, induction of PPO cascade	DL2	c-type lectin 2	NP_001014489.1	XP_016663197.1 XP_001950803.2 XP_001945032.2	QBOQ01000461.1 (1 × 10^−52^) QBOQ01000461.1 (3 × 10^−53^) QBOQ01000466.1 (4 × 10^−24^)
bacterial recognition, induction of PPO cascade	DL3	c-type lectin 3	NP_001014490.1	XP_016663197.1 XP_001950803.2	QBOQ01000461.1 (1 × 10^−52^) QBOQ01000461.1 (3 × 10^−53^)
several roles have been hypothesized	galectin	galectin 9	ADZ99399.1	XP_001943769.2	not found
mark pathogens for phagocytosis	Tep1	thioester containing protein 1	CAB87807.1	XP_029348718.1	QBOQ01000202.1 (2 × 10^−38^)
mark pathogens for phagocytosis	Tep2	thioester containing protein 2	CAB87808.1	XP_029348718.1	QBOQ01000202.1 (2 × 10^−38^)
mark pathogens for phagocytosis	Tep3	thioester containing protein 3	AAL39195.1	XP_029348714.1	QBOQ01000202.1 (2 × 10^−38^)
mark pathogens for phagocytosis	Tep4	thioester containing protein 4	NP_523603.2	XP_029348718.1	QBOQ01000202.1 (2 × 10^−38^)
bacterial and fungal recognition	pes	peste, scavenger receptor class b	AHN54246.1	XP_029341846.1	QBOQ01001218.1 (2 × 10^−33^)
bacterial and fungal recognition	crq	croquemort	AAF51494.1	XP_001944867.2	QBOQ01000024.1 (1 × 10^−35^)
bacterial and fungal recognition	drpr	draper	NP_477450.1	XP_001942552.2	QBOQ01001915.1 (2 × 10^−29^)
bind to lipoproteins and bacteria	sr-CI	scavenger receptor class c, type i	AAW79470.1	not found	not found
bind to lipoproteins and bacteria	sr-CII	scavenger receptor class c, type ii	AAF58551.1	not found	not found
bind to lipoproteins and bacteria	sr-CIII	scavenger receptor class c, type iii	AAF37564.1	not found	not found
bind to lipoproteins and bacteria	sr-CIV	scavenger receptor class c, type iv	AAF51092.1	not found	not found
receptor in phagocytosis and microbial binding	eater	eater	AAF56664.5	not found	not found
receptor in phagocytosis and microbial binding	nim-C1	Nimrod c1	AAF53364.2	not found	not found

^1^ Alternative names are separated by commas. ^2^ E-values refer to *A. pisum* proteins, if present, or *D. melanogaster* if the gene was not found in *A. pisum*.

**Table 2 ijms-25-04922-t002:** Immune genes of *Ericerus pela* involved in signaling. Genes not found in *E. pela* are colored in red.

Role in Insect Immunity	Gene Symbol	Gene Name ^1^	*D. melanogaster* NCBI Protein ID	*A. pisum* NCBI Protein ID	Best Matches (E-Value) ^2^
Toll pathway	spz1-1	spätzle 1B	NP_733188.1	NP_001153589	QBOQ01001621.1 (7 × 10^−13^)
Toll pathway	spz1-2	spätzle 1Bii	NP_001138116.1	NP_001153590	QBOQ01001132.1 (9 × 10^−8^)
Toll pathway	spz2	spätzle 2, neurotrophin 1	NP_001261417.1	XP_001948459.1	QBOQ01001810.1 (4 × 10^−20^)
Toll pathway	spz3	spätzle 3	NP_609160.2	XP_029341989.1	QBOQ01001601.1 (2 × 10^−21^) QBOQ01001203.1 (2 × 10^−14^) QBOQ01000537.1 (9 × 10^−13^)
Toll pathway	spz4	spätzle 4	NP_609504.2	NP_001153592	QBOQ01000537.1 (2 × 10^−28^) QBOQ01001203.1 (4 × 10^−28^) QBOQ01001601.1 (2 × 10^−14^)
Toll pathway	Spz5	spätzle 5	NP_647753.1	XP_001947495.2	QBOQ01001431.1 (2 × 10^−17^)
Toll pathway	spz6	spätzle 6	NP_611961.1	XP_001944046	QBOQ01001423.1 (6 × 10^−58^)
Toll pathway	Toll-1	protein Toll	NP_524518.1	XP_008182102.1	QBOQ01001036.1 (3 × 10^−57^) QBOQ01001364.1 (4 × 10^−54^) QBOQ01000985.1 (8 × 10^−48^)
Toll pathway	Toll-1	protein Toll	NP_524518.1	XP_001942733.2	QBOQ01001036.1 (6 × 10^−54^) QBOQ01001364.1 (7 × 10^−49^) QBOQ01000985.1 (2 × 10^−38^)
Toll pathway	18w	18 wheeler, Toll-2	NP_476814.1	XP_001946943.2	QBOQ01000059.1 (0.0) QBOQ01000985.1 (0.0) QBOQ01000048.1 (0.0)
Toll pathway	Toll-6	Toll-6	NP_001246766.1	XP_001947324.1	QBOQ01000985.1 (0.0) QBOQ01000059.1 (0.0) QBOQ01000048.1 (0.0)
Toll pathway	Toll-6	Toll-6	NP_001246766.1	XP_003248960.1	QBOQ01000048.1 (0.0) QBOQ01000985.1 (0.0) QBOQ01000059.1 (0.0)
Toll pathway	Toll-7	Toll-7	NP_523797.1	XP_001946943_2	QBOQ01000059.1 (0.0) QBOQ01000985.1 (0.0) QBOQ01000048.1 (0.0)
Toll pathway	Tollo	Tollo, Toll-8	NP_524757.1	XP_001948566.1	QBOQ01000985.1 (0.0) QBOQ01000059.1 (0.0) QBOQ01000048.1 (0.0)
Toll pathway	tub	tube, interleukin-1 receptor-associated kinase 4	NP_001189164.1	BAH72505.1	QBOQ01000327.1 (8 × 10^−15^)
Toll pathway	Myd88	myeloid differentiation primary response gene	AAF58953.1	XP_001948320.2	not found
Toll pathway	pll	pelle	AAF56686.1	XP_029346632.1	QBOQ01000327.1 (2 × 10^−33^) QBOQ01002061.1 (3 × 10^−12^) QBOQ01001518.1 (3 × 10^−11^)
Toll pathway	cact	cactus	AAN10936.1	NP_001156668.1	not found
Toll pathway	cactin	cactin	NP_523422.4	XP_001952287.2	QBOQ01001452.1 (1 × 10^−88^)
Toll pathway	Pli	pellino	NP_524466.1	XP_001946282.3	QBOQ01001351.1 (4 × 10^−29^)
Toll pathway	Traf1, Traf4	TNF-receptor-associated factor 1	AAD34346.1	XP_001948355.1	QBOQ01000448.1 (3 × 10^−67^)
Toll pathway	Traf2, Traf6	TNF-receptor-associated factor 2	AAF46338.1	XP_029347356.1	QBOQ01001366.1 (1 × 10^−17^)
Toll pathway	Traf3, Traf-like	TNF-receptor-associated factor 3	NP_727976.1	not found	not found
Toll pathway	dl	dorsal	AAF53611.1	XP_001949498.2	QBOQ01000587.1 (1 × 10^−68^)
Toll pathway	Dif	dorsal-related immunity factor, embryonic polarity protein	NP_523589.2	XP_001949498.2	QBOQ01000587.1 (1 × 10^−68^)
Jak/stat pathway	dome	domeless 1, interleukine JAK/STAT receptor	CAD12503.1	XP_029341085.1	QBOQ01000913.1 (1 × 10^−114^)
Jak/stat pathway	dome2	domeless 2	Not found	XP_029341036.1	QBOQ01000913.1 (6 × 10^−111^)
Jak/stat pathway	hops, jak	hopscotch, Janus kinase	NP_511119.2	XP_008188128.1	QBOQ01001628.1 (2 × 10^−29^) QBOQ01002061.1 (4 × 10^−28^) QBOQ01000952.1 (6 × 10^−24^)
Jak/stat pathway	Stat92E	signal-transducer and activator of transcription, marelle	AAX33462.1	XP_008188159.1	QBOQ01001541.1 (5 × 10^−44^) QBOQ01000405.1 (1 × 10^−42^)
Jak/stat pathway	upd1	unpaired 1	NP_525095.2	not found	not found
Jak/stat pathway	upd2	unpaired 2	NP_001356882.1	not found	not found
Jak/stat pathway	upd3	unpaired 3	NP_001097014.1	not found	not found
Imd pathway	imd	immune deficiency	NP_573394.1	not found	not found
Imd pathway	dFadd	dFadd	NP_651006.1	not found	not found
Imd pathway	Dredd	death related ced-3, caspase-1	NP_477249.3	XP_029344969.1	QBOQ01001252.1 (4 × 10^−61^)
Imd pathway	Rel	Relish	NP_477094.1	not found	not found
Imd pathway	Tab2	TAK1-associated binding protein 2	NP_611408.2	XP_003244590.1	QBOQ01001392.1 (7 × 10^−8^)
Imd pathway	Tak1	TGF-β activated kinase 1	AAF50895.1	XP_029347425.1	QBOQ01001920.1 (3 × 10^−35^) QBOQ01001518.1 (6 × 10^−29^) QBOQ01000779.1 (1 × 10^−17^)
Imd pathway	key	kenny	NP_523856.2	not found	not found
Imd pathway	Diap2	death-associated inhibitor of apoptosis 2	NP_477127.1	XP_016661891.1	QBOQ01002166.1 (3 × 10^−20^) QBOQ01001565.1 (2 × 10^−12^) QBOQ01001600.1 (4 × 10^−12^)
Imd pathway	ird5	immune response deficiency 5, IK-β, IKKB, I-kappaB kinase beta	NP_524751.3	XP_001946184.1	QBOQ01000860.1 (0.0)
Jnk pathway	hep	hemipterous	NP_727661.1	XP_008180171.1	QBOQ01001476.1 (1 × 10^−126^)
Jnk pathway	bsk	basket	P92208.1	XP_001945460.2	QBOQ01001118.1 (6 × 10^−45^) QBOQ01001795.1 (4 × 10^−40^) QBOQ01001783.1 (4 × 10^−26^)
Jnk pathway	Jra	Jun-related antigen	AAF58845.1	XP_001947556.1	QBOQ01001648.1 (8 × 10^−16^)
Jnk pathway	kay	kayak	NP_001027579.1	XP_016663984.1	not found
Jnk pathway	egr	Eiger	AAF58848.2	XP_008178962.1	QBOQ01000141.1 (7 × 10^−5^)

^1^ Alternative names are separated by commas. ^2^ E-values refer to *A. pisum* proteins, if present, or *D. melanogaster* if the gene was not found in *A. pisum*.

**Table 3 ijms-25-04922-t003:** Immune genes of *Ericerus pela* involved in response. Genes not found in *E. pela* are colored in red.

Role in Insect Immunity	Gene Symbol	Gene Name ^1^	*D. melanogaster* NCBI Protein ID	*A. pisum* NCBI Protein ID	Best Matches (E-Value) ^2^
antimicrobial peptide	Att	attacin	NP_523745.1	not found	not found
antimicrobial peptide	Cec	cecropin	C0HKQ7.1	not found	not found
antimicrobial peptide	Def	defensin	ANY27112.1	not found	not found
antimicrobial peptide	Dro	drosocin	XP_016946682.1	not found	not found
antimicrobial peptide	Mtk	metchnikowin	AAO72489.1	not found	not found
antimicrobial peptide		andropin	P21663.1	not found	not found
antimicrobial peptide		diptericin	QER92349.1	not found	not found
antimicrobial peptide	Drs	drosomycin	ANY27466.1	not found	not found
antimicrobial peptide		holotricin	XP_051861657.1	not found	not found
antimicrobial peptide		bomanin	A1ZB62.1	not found	not found
antimicrobial	LOC100164856	thaumatin-like protein	not found	XP_001942718.2	not found
antimicrobial	LOC100160062	thaumatin-like protein 1b	not found	XP_001942572.1	not found
antimicrobial	LOC100570639	thaumatin-like protein 1	not found	XP_003248856.4	not found
antimicrobial	LOC100162111	uncharacterized LOC100162111, thaumatin family	not found	NP_001155516	not found
antimicrobial	LOC100168942	TLP-PA-domain protein	not found	NP_001156304.1	not found
antimicrobial	LOC100169496	pathogenesis-related protein 5-like	not found	NP_001313585.1	not found
microbial degradation	LysX	lysozyme X, i-type	CAL85493.1	not found	not found
microbial degradation	LysB	lysozyme B, i-type	NP_001261245.1	not found	not found
microbial degradation	LysP	lysozyme, i-type	NP_476828.1	not found	not found
microbial degradation	LysC	lysozyme	CAA80228	not found	not found
microbial degradation	LysD	lysozyme	NP_476823.1	not found	not found
microbial degradation	LysE	lysozyme	NP_476827.2	not found	not found
microbial degradation	LysS	lysozyme	NP_476829.1	not found	not found
microbial degradation	lysozyme, i-type	LOC100167742, lysozyme	ACD99447.1	XP_001949318.2	QBOQ01000327.1 (6 × 10^−19^) QBOQ01001156.1 (5 × 10^−18^) QBOQ01000040.1 (7 × 10^−18^)
microbial degradation	lysozyme, i-type	LOC100168424, destabilase	NP_611164.3	NP_001156290.1	QBOQ01002128.1 (5 × 10^−17^) QBOQ01000327.1 (6 × 10^−8^) QBOQ01000040.1 (9 × 10^−8^)
microbial degradation	lysozyme, i-type	LOC100160909, destabilase	NP_611163.2	NP_001155465.1	QBOQ01000040.1 (3 × 10^−11^) QBOQ01002128.1 (8 × 10^−8^)
fungal degradation	Cht2	chitinase-like protein 2, mucin	NP_001261282.1	XP_016663378.1	QBOQ01000205.1 (1 × 10^−40^) QBOQ01000535.1 (1 × 10^−19^) QBOQ01001282.1 (3 × 10^−19^)
fungal degradation	Cht4	chitinase-like protein 4, flocculation protein	NP_524962.2	XP_029343203.1	QBOQ01001282.1 (3 × 10^−40^) QBOQ01000205.1 (3 × 10^−20^) QBOQ01000535.1 (5 × 10^−19^)
fungal degradation	Cht5	chitinase-like protein 5, endochitinase	NP_650314.1	XP_008181779.1	QBOQ01000410.1 (3 × 10^−47^) QBOQ01000205.1 (3 × 10^−16^) QBOQ01001282.1 (5 × 10^−21^)
fungal degradation	Cht6	chitinase-like protein 6, flocculation protein	NP_001245602.1	XP_029343203.1	QBOQ01001282.1 (3 × 10^−40^) QBOQ01000205.1 (3 × 10^−20^) QBOQ01000535.1 (5 × 10^−19^)
fungal degradation	Cht7	chitinase-like protein 7, chitinase 10	NP_647768.3	XP_001950380.1	QBOQ01000535.1 (9 × 10^−87^) QBOQ01000062.1 (2 × 10^−61^) QBOQ01001282.1 (3 × 10^−24^)
fungal degradation	Cht7	chitinase 3-like, LOC100169240	NP_647768.3	XP_008182858.1	QBOQ01002026.1 (6 × 10^−19^) QBOQ01000535.1 (8 × 10^−14^) QBOQ01000205.1 (4 × 10^−8^)
fungal degradation	Cht6	LOC100162732	NP_001245599.1	XP_001945470.2	QBOQ01001292.1 (2 × 10^−7^)
fungal degradation	idgf6	idgf	NP_001286499.1	NP_001162142.1	QBOQ01000713.1 (3 × 10^−55^)
prophenoloxidase response	PPO1	prophenoloxidase 1	NP_476812.1	XP_001949307.1	QBOQ01000687.1 (5 × 10^−45^) QBOQ01000496.1 (8 × 10^−33^) QBOQ01002014.1 (3 × 10^−27^)
prophenoloxidase response	PPO2	prophenoloxidase 2	NP_610443.1	XP_001951137.1	QBOQ01000687.1 (2 × 10^−43^) QBOQ01000496.1 (2 × 10^−37^) QBOQ01002014.1 (5 × 10^−27^)
phenoloxidase activation	PAF2, PPAF2	phenoloxidase-activating factor 2	AAO24923.1	XP_003244500.1	QBOQ01000870.1 (3 × 10^−13^) QBOQ01001093.1 (2 × 10^−12^) QBOQ01001364.1 (2 × 10^−9^)
phenoloxidase activation	PAF2, PPAF2	phenoloxidase-activating factor 2	AAO24923.1	XP_001952301.1	QBOQ01000870.1 (2 × 10^−24^) QBOQ01001093.1 (2 × 10^−28^) QBOQ01002068.1 (1 × 10^−15^)
phenoloxidase activation	SP	serine protease-like precursor	NP_001097766.1	NP_001155379.1	QBOQ01000877.1 (2 × 10^−24^) QBOQ01000234.1 (2 × 10^−23^) QBOQ01001364.1 (6 × 10^−8^)
cell aggregation	Hmct, hemolectin	hemocytin	NP_001261809.1	XP_001952865.2	QBOQ01002094.1 (3 × 10^−22^)
production of nitric oxide, a toxic gas	Nos	nitric oxide synthase	NP_001027243.2	XP_029343919.1	QBOQ01001094.1 (4 × 10^−29^)
peptidoglycan degradation	ldca	putative LD carboxypeptidase	not found	XP_029341985.1	QBOQ01000175.1 (2 × 10^−43^)

^1^ Alternative names are separated by commas. ^2^ E-values refer to *A. pisum* proteins, if present, or *D. melanogaster* if the gene was not found in *A. pisum*.

## Data Availability

All fasta sequences used and alignments are available from Zenodo at https://doi.org/10.5281/zenodo.10864136.

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
