# Peer review of "Immune Gene Repertoire of Soft Scale Insects (Hemiptera: Coccidae)"

_ijms, 2024, doi:10.3390/ijms25094922_

Round 1
Reviewer 1 Report
Comments and Suggestions for Authors
Please see the attached file.

Author Response
Please, check our reply in the attached file.

Reviewer 2 Report
Comments and Suggestions for Authors
Dear Authors,
I read carefully your submitted article IJMS-2965955, and I consider it a quite interesting and advanced contribution into knowledge of immune gene system of the species belonging to Coccidae family. Each section of the mns has been clearly descibed, i.e. Introduction , Results, Discussion and Conclusions. All the referencies recorded in the text are appropriate and properly cited into sections. However, in my review , I saw that some parts of the M&M chapter need to be re-written , in order to clarify the use of the softwares in the main steps . In particular: sentence from line 459 to 460 needs explanation for "e-value for threshold set at 1e-5 ". Also, please, a more detailed explaination of the use of software getorf, i.e. from line 463 to 467 , in order to reconstruct of Coccidae immune gene phylogeny. Moreover, I suggest to re-write the period from line 468 to 474 to clarify how the use of D.melanogaster proteins produced putative homologous sequences. Clarification also for the sentence from line 472 to 474. These modifications through more details, are important to improve the quality presentation of the paper, because a simple record of the literature reference is not enough to make understand to readers the modes of operation. Moreover , clear details can favour the use of such methods by other researchers.
Sincerely
Author Response

(The authors gave the same response as above.)

Round 2
Reviewer 1 Report
Comments and Suggestions for Authors
Given the quite interesting contribution into knowledge of innate immunity of the species belonging to Coccidae family, I believe that this review is suitable for publication in IJMS.
Reviewer 2 Report
Comments and Suggestions for Authors
Dear Authors,
I just read the new version of your mns ijms-2965955 v2, and I am agree with . In particular, I appreciated the re-writing of the M&M section. As I suggested in the previous review.
Sincerely